# CanaryExp: A Canary-Sensitive Automatic Exploitability Evaluation Solution for Vulnerabilities in Binary Programs

Hui Huang [1,2], Yuliang Lu [1,2], Kailong Zhu [1,2,*] and Jun Zhao [1,2]

1 College of Electronic Engineering, National University of Defense Technology, Hefei 230037, China; huanghui17@nudt.edu.cn (H.H.); luyuliang@nudt.edu.cn (Y.L.); zhaojun17@nudt.edu.cn (J.Z.)
2 Anhui Province Key Laboratory of Cyberspace Security Situation Awareness and Evaluation, Hefei 230037, China
* Correspondence: zhukailong@nudt.edu.cn

**Abstract:** We propose CanaryExp, an exploitability evaluation solution for vulnerabilities among binary programs protected by StackGuard. CanaryExp devises three novel techniques, namely canary leakage proof of concept generation, canary leaking analysis time exploitation, and dynamic canary-relocation-based exploitability evaluation. The canary leakage proof of concept input generation mechanism first traces the target program's execution, transforming the execution state into some canary leaking state, from which some canary leaking input is derived. This input can be deemed as proof that some vulnerability that can lead to canary leakage exists. The canary leaking analysis time exploit generation then performs incremental analysis based on the canary leaking input, crafting analysis time exploit that can complete vulnerability exploitation in the analysis time environment. Based on the analysis time exploit, the dynamic canary-relocation-based exploitability evaluation component collects the necessary metadata, on which an exploitation session is automatically constructed that can not only leak the runtime canary and relocate it in the input stream but also evaluate the exploitability of the desired vulnerability. Using a benchmark containing six test programs, eight challenges from some network challenging events and four real-world applications, we demonstrate that CanaryExp can generate canary leaking samples more effectively than existing test case generation methods and automatically evaluate the exploitability for vulnerabilities among programs where the StackGuard protection mechanism is deployed.

**Keywords:** vulnerability; canary; information leak; automatic exploit generation





## 1. Introduction

With lots of vulnerabilities being discovered every day by security analysts all over the world, how to evaluate the exploitability of vulnerabilities in a more automatic and effective way is becoming a critical problem for security researchers. In recent years, automatic exploit generation (abbr. AEG) has become the mainstream topic in this researching direction. Many solutions including AEG [1], CRAX [2], REX [3], and Mayhem [4] have been proposed, discovering vulnerabilities and automatically generating exploits when possible, for source code and binary, respectively. In general, these methods often first analyze vulnerabilities in detail along the trace of a proof of concept input (abbr. POC). Note that a proof of concept input is generally an input sample that can exhibit the existence of some vulnerability. Through the trace analysis, the critical execution state where the desired vulnerability is triggered is automatically derived. Based on the exact state, further exploration is performed in order to find a subsequent state that can be judged as some exploitable state. If some exploitable state is actually discovered, the path reachability constraint, vulnerability triggering constraint and exploit construction constraint would separately be collected, with the conjunction of these constraints solved using constraint solvers following the satisfiability modulo theories (abbr. SMT) [5], through which the exploit input is finally generated.

While proven effective in addressing classical vulnerabilities in binary programs, including stack overflow [6,7], heap overflow [8–11], and use-after-free [12–14], the method remains adaptable under circumstances where exploitation mitigation options, including non-executable stack (abbr. NX) [15], address space layout randomization (abbr. ASLR) [16], and SafeSEH [17], are deployed. However, for vulnerable programs where the StackGuard [18] exploitation mitigation option is turned on and it is necessary to actively bypass this mitigation in the exploitability evaluation process, current methods generally fail in verifying the exploitability of the desired vulnerability.

StackGuard prevents the return address from being changed by overflow vulnerabilities by inserting a 'canary' on the way from the local buffer variable to the return address stored on the stack. When the control returns after a function body's execution, it checks whether the canary is not altered by comparing it with its original copy saved somewhere else, before jumping to the function's return address. If the canary is determined to be overwritten, StackGuard terminates the execution immediately, therefore protecting the program from being attacked through buffer overflow vulnerabilities. Currently, this mechanism is taken as a default option in modern compilation systems [19,20], and is widely adopted in real-world applications. To complete vulnerability evaluation under this condition, human analysts generally have to leak the canary first, then place the leaked canary at the exact location in the exploitation payload, hijacking the control flow through stack smashing while bypassing the runtime canary check enforced by StackGuard as if the desired canary has not been overwritten yet. We find that three key challenges should be resolved to make current AEG systems adaptable to this issue. These challenges are listed below.

*Challenge 1: canary leakage event forgery.* As StackGuard-protected programs would not explicitly manifest the canary value to the external analyst, POC input generally smashes the stack structure completely, leading the execution to termination. Therefore, it is the AEG system's duty to forge a new canary leaking event from scratch, establishing the first step towards complete exploitation of StackGuard-protected programs.

*Challenge 2: sanity check bypassing input generation.* Obtainment of a canary leaking input is not the termination. AEG system should explore the path space alongside this input to further discover some exploitable state, only on which can we derive a complete exploit input. However, as for most current test case generation techniques including fuzzing [21–23], symbolic execution [24,25], hybrid fuzzing [26,27], etc., the sanity check performed by StackGuard would definitely block exploitable state from being discovered as the stack structure is generally smashed. A sanity check bypassing input generation scheme should be proposed to help the AEG system tackle the issue in an effective way.

*Challenge 3: dynamic canary relocation.* Current AEG systems often generate exploits under the assumption that the analysis environment and the runtime environment match exactly the same. However, this is not the case for StackGuard-protected programs, as the canary value it adopted differs between each bootstrap of the environment. It is necessary to equip an extra canary relocation device to current AEG systems to fix the critical environmental gap.

*Our Solution.* We present CanaryExp, a canary-sensitive automatic vulnerability exploitability evaluation system to address the above challenges. Given a binary program with some memory corruption vulnerability and a POC input that can trigger some vulnerability but cannot hijack the control flow due to failure on canary checking performed by StackGuard, CanaryExp firstly monitors the execution of the POC input, generating input samples that can leak the desired canary at runtime. Based on this canary leaking input, it then generates input samples that can not only leak the canary, but also trigger some new vulnerability afterwards. We define these input samples as incremental proof of concept (abbr. incremental POC) throughout this paper. The incremental POC is then utilized by an exploitation technique applier generating exploit input that fits the analysis time environment. An extra canary relocation process is deployed at last, making the analysis

time exploit adaptable to the runtime environment, providing a completely automatic exploitability evaluation workflow for security analysts.

We have constructed a prototype system of CanaryExp based on the open source binary analysis platform angr [28], coverage guided fuzzing engine AFL++ [29] and whole-system symbolic executor S2E [30], and evaluated it on six test programs, eight CTF challenges and four real-world applications. The results of experiments show that (1) CanaryExp can generate canary leaking input and incremental POC able to both leak the canary and trigger some vulnerability more efficiently than current test case generation systems, and (2) CanaryExp can automatically and effectively evaluate the exploitability of the desired vulnerabilities existing in the 18 vulnerable programs that are protected by StackGuard.

The main contributions of this paper are listed below:

- We propose a novel canary leakage forge method. By performing output buffer analysis and canary recognition through dynamic state monitoring alongside the execution of a POC input, the overflowed canary in the output buffer is recovered with a canary leaking execution state derived on which canary leaking input can be automatically calculated.
- We propose a canary leaking analysis time exploit generation method. By exploring state space alongside test cases that are able to leak the desired canary and disabling StackGuard's sanity check during exploitable state derivation, the analysis time exploit is generated providing a data template for later automatic construction of exploitability evaluation scheme.
- We propose a dynamic canary-relocation-based exploitability evaluation mechanism. By collecting metadata associated with the runtime canary and the IO sequence implied by the analysis time exploit, an exploitation session is dynamically constructed, during which real-time extraction and relocation of canary value are performed, providing automatic exploitability evaluation workflow for vulnerabilities in StackGuard-protected programs.
- We have implemented a prototype system of CanaryExp and proved its effectiveness in automatic exploitability evaluation for vulnerabilities in StackGuard-protected programs.

## 2. Motivation Example

In this section, we first introduce the manual exploitation process generally taken by security analysts for a simple vulnerable StackGuard-protected program, then discuss the limitations of current AEG systems in automatic exploitability evaluation for this vulnerability, and finally propose the insights we believe current AEG systems should follow to tackle this issue.

### 2.1. The Vulnerability and StackGuard Mechanism

For ease of illustration, we compile the example program shown in Figure 1a by executing the command 'gcc demo.c -m32 -static -fstack-protector -o demo' under Linux. After compilation, we obtain a 32-bit executable image that is statically linked in ELF format, with StackGuard option turned on.

As can be seen in the example, the protection mechanism provided by StackGuard is composed of 2 key components—one for canary storage and the other a sanity check. The canary storage phase is shown by ① in Figure 1b, consisting of two store instructions copying memory content located by logical address 'gs:0x14' (which is the defacto runtime canary initialized by the glibc runtime environment) to a memory cell in the stack frame. After the canary storage operation is completed, the layout of function func's stack frame becomes the one shown in Figure 1c, from which we can see that the stored runtime canary resides at an address that is both lower than the memory cells that containing the caller's EBP and call site address and higher than the address of the variable 'buffer' on the stack.

The sanity check phase is shown by ② in Figure 1b. It is located before the return point of function func. The sanity check firstly fetches the value stored at virtual address

'ebp-0xc', then checks whether it equals the original canary through an exclusive OR operation. If the two values are the same, the control flow would return normally; otherwise, function __stack_chk_fail_local would be invoked (shown by ③ in Figure 1), immediately terminating the execution.

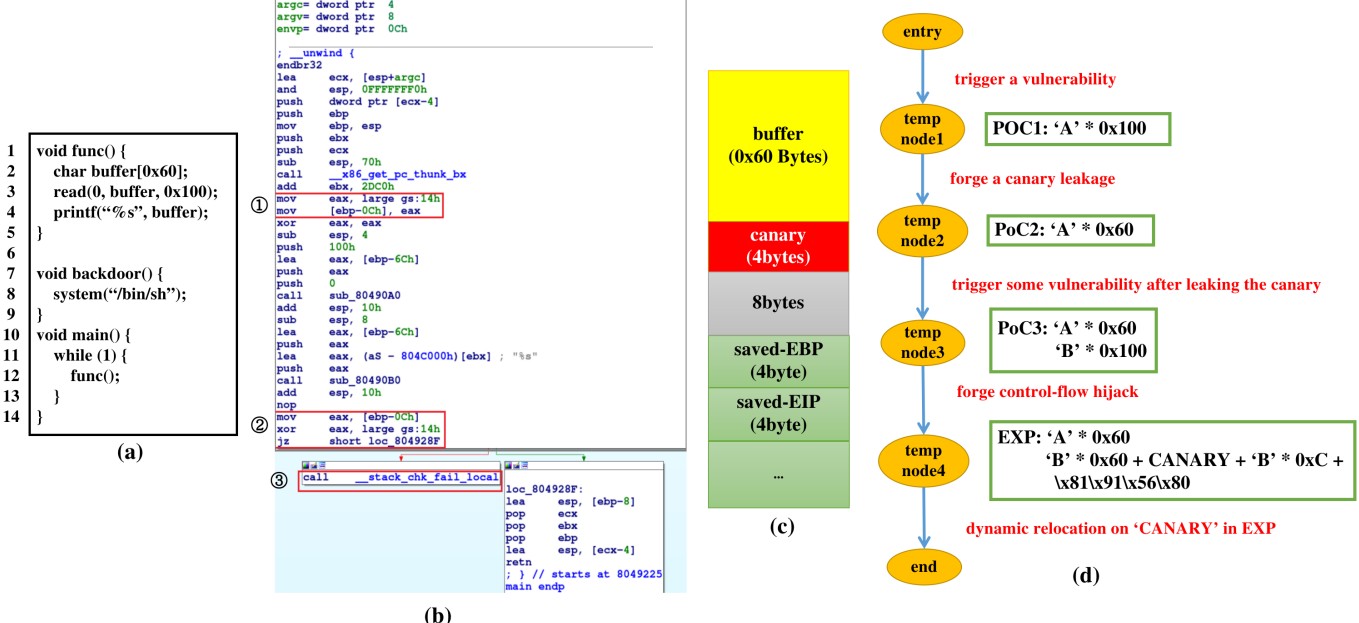

**Figure 1.** An example of exploitation of vulnerability in a StackGuard-protected program. (**a**) Source code of the example. (**b**) Disassembly view of function *func* in this example. This view is generated by the well-known interactive disassembler IDA Pro. (**c**) Stack frame of *func*. (**d**) Steps we believe AEG systems should take to handle this case.

We can see in line 3 in Figure 1a that call on read occurs. Due to this call, a stack overflow vulnerability exists as at most 0x100 bytes can be read from the external environment and filled into a 0x60-sized buffer. For programs not protected by StackGuard, a simple input containing 0x100 'A' would directly hijack the control flow, facilitating current AEG systems generating effective exploit. However, it is not the case for this example.

As the canary to be checked lays in the stack region between the input buffer and saved return address, overwriting the return address through stack overflow on the local variable buffer would inevitably overwrite the stored canary. Unless the stored canary is overwritten with the exact value StackGuard adopts at runtime, which is generally impossible for current test case generation methods, the execution would generally directly terminate as the sanity check fails in this case.

### 2.2. Inspirations on AEG

Traditional AEG systems generally follow a one-step strategy. Assuming some exploitable state can be explicitly derived through trace analysis of the provided POC input, they try to deduce an exploitable state from the vulnerable state manifested by the POC input. As stack smashing inputs cannot bypass the sanity check imposed by StackGuard, this strategy no longer holds under this circumstance.

We believe a multi-step evolutionary policy is needed to handle this issue. Therefore, we propose the multi-step exploitation procedure shown in Figure 1d, in which five steps are required to complete the automatic exploitability evaluation process of the example program.

In the first step, we get a simple stack smashing input through available test case generation techniques. Though the input manifests already the existence of a stack overflow

vulnerability, it can not pass the protection imposed by StackGuard. We need to forge some canary leakage event.

Through analyzing the stack frame layout and the execution trace designated by POC1, we can transform the original stack smashing event into a canary leaking event, with a new test case POC2 containing 0x60 'A's generated.

As POC2 only manifests the existence of stack canary leakage, we take step 3 performing state space exploration alongside the prefixed path designated by POC2. This time, a new test case POC3 is generated, which can not only leak the desired canary, but also cause the execution to reach some exploitable state. It is worth noting that as the sanity check performed by StackGuard may block the control flow hijack state being discovered, it should be temporarily blocked in this phase.

The fourth step performs exploit generation of the exploitable state implied by POC3, generating a new input EXP that can be used to evaluate the exploitability of the target vulnerability in the analysis time environment. The fifth step then performs an extra relocation process of EXP. By filling CANARY with the leaked canary value in runtime, a complete automatic exploitation process is finally accomplished.

## 3. Overview of CanaryExp

We propose CanaryExp, an automatic vulnerability exploitability evaluation solution for vulnerable binary programs where the StackGuard mechanism is deployed. As shown in Figure 2, CanaryExp consists of three components: separate canary leakage POC generation, canary leaking analysis time exploit generation, and dynamic canary-relocation-based exploitability evaluation.

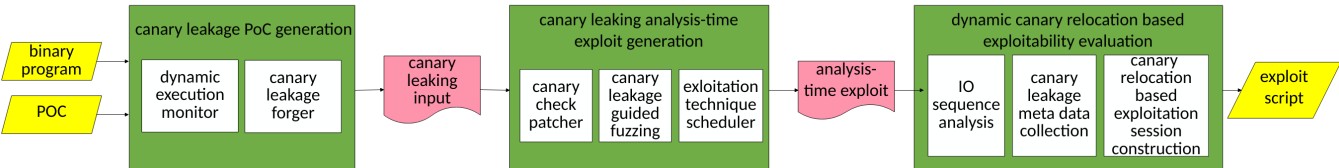

**Figure 2.** Overview of CanaryExp.

CanaryExp works in the multi-step automatic exploit generation paradigm discussed in Section 2.2. Given a binary program and some POC input, CanaryExp first employs the canary leakage POC generation engine generating input sample that can leak the desired canary value. This canary leaking input is then fed to the canary leaking analysis time exploit generation engine, which firstly generates some incremental POC input sample that can trigger some new vulnerability after leaking the desired canary through canary-leakage-guided fuzzing, then based on the generated sample, generates analysis time exploit through enumerating possible exploitation techniques. The analysis time exploit is then directed to the dynamic canary-relocation-based exploitation evaluation engine, which generates an exploit script that can perform dynamic relocation of the canary part of the analysis time exploit.

## 4. Canary Leakage POC Generation

As depicted in Figure 2, the canary leakage POC generation engine firstly traces along the execution specified by a vulnerability POC input, then constructs a canary leakage forger atop it. This section illustrates the details of this component.

### 4.1. Dynamic Execution Monitor

The dynamic execution monitor engine is built upon a dynamic binary translation-based software virtual machine [31]. It performs concolic execution [26,27] alongside the execution trace specified by a POC input. Under this background, the execution state of the program can be represented by a quintuple $state :< \Sigma, \mu, \Delta, \Pi, \delta >$, among which

$\Sigma : \mathbb{Z} \to$ **Instructions** represents the mapping between virtual addresses and program instructions, $\mu : \mathbb{Z} \to \mathbb{Z} \bigcup$ **SYMBOLICS** represents the definition state of memory cells, $\Delta :$ **REG_NAMES** $\to \mathbb{Z} \bigcup$ **SYMBOLICS** represents the definition state of registers, $\Pi \in$ **SYMBOLICS** represents the path constraint, $\delta : \mathbb{Z} \to$ **SYMBOLICS** records the symbolic variables introduced on each read offset from the external input.

During the sequential execution of each instruction in the execution trace, the program state is constantly influenced by the operation semantics implied by the executed instructions [32]. We define the tuple $ioctx :< \gamma, v, \sigma >$ to record the output context during the execution, among which $\gamma$ represents the anticipated information leakage channel (this may be the standard output, network communication endpoint, etc.), $v$ represents the length of data written to $\gamma$, and $\sigma \in \mathbb{Z}$ holds the canary. We place hooks at output API functions' returning sites, recording the update of $v$.

The dynamic execution monitor engine also exposes program instrumentation interfaces during the dynamic binary translation process. These interfaces provide a generic hooking mechanism of semantics at both the instruction execution level and the guest operating system level. Hooks at the instruction execution level are implemented by inserting trampolines in the translation and decode phase of the guest instruction in the dynamic binary translation process. As for hooks at the guest operating system level, a kernel mode driver is customized. The driver captures the process-related events in the guest operating system and makes immediate notification to the execution monitor so as to invoke the corresponding instrumentation code. These interfaces are listed in Table 1.

**Table 1.** Instrumentation interfaces.

| Name | Level | Hooking Semantics |
|---|---|---|
| onMemRead | instruction execution | A memory read operation is performed |
| onMemWrite | instruction execution | A memory write operation is performed |
| onRegRead | instruction execution | A register read operation is performed |
| onRegWrite | instruction execution | A register write operation is performed |
| onInstructionStart | instruction execution | An instruction begins to execute |
| onInstructionEnd | instruction execution | An instruction finishes its execution |
| onBlockStart | instruction execution | A basic block begins to execute |
| onBlockEnd | instruction execution | A basic block finishes its execution |
| onFunctionStart | instruction execution | A function begins to execute |
| onFunctionRet | instruction execution | A function finishes its execution |
| onModuleLoad | operating system | A module is loaded into the address space of the target process |
| onModuleUnload | operating system | A module is unloaded into the address space of the target process |

*4.2. Canary Leakage Forger*

The canary leakage forger aims to construct an execution state that can leak the desired canary. It consists of two components, namely activation record analysis and output stream analysis.

Each time a procedure in the target program is invoked, a new activation record is allocated on the stack region, keeping storage for the local variables, caller's return address, caller's stack base pointer, and the runtime canary imposed by StackGuard. When the control flow returns to the caller from this function, the activation record is removed.

The activation record analysis component keeps track of the structure information for all currently alive activation records in $\theta : \{< base, top, canary > | base \in \mathbb{Z}, top \in \mathbb{Z}, canary \in \mathbb{Z}\}$, among which a single triple $< base, top, canary >$ represents a single activation record, with *base* and *top* denoting the base address and top address of the activation record region on the stack, *canary* denoting the address on the stack frame where the stored

canary resides in. By installing hooks on instrumentation interfaces listed in Table 1, these values are calculated dynamically.

The output stream analysis monitors the output events that happened along the execution trace by arranging hooks at the entry points of output API functions, trying to derive an execution state that can leak the desired canary. It completes the state derivation through two methods: one for activation record restoration and the other for symbolic output buffer manipulation.

Formula (1) demonstrates an example of operation semantics of activation record restoration at the entry point of the Posix write function. Note that we describe the operation semantics in the same way as [32].

When the write function is invoked in the form $write(fd, buf, sz)$, the activation record restoration method firstly checks if fd points to the desired information leaking channel. If so, it locates the activation record sf where buf resides in, and ensures that memory region $\mu[buf, sf.canary]$ is filled with symbolic variables. It then checks the mathematical relationship among the symbolic variables in memory region $\mu[buf, sf.canary]$. If some affine relationship is discovered [33], the activation record restoration method truncates the original symbolic content sequence $\delta$ to $\delta|_{\{0,...,n\}}$, where $n$ is the maximum index of symbolic variables in $\mu[buf, sf.canary - 1]$. In this way, the canary leakage state $< \Sigma, \mu, \Delta, \Pi, \delta|_{\{0,...,n\}} >$ is immediately derived.

$$
\begin{gathered}
\Sigma[\Delta["pc"]] = write(fd, buf, sz) \wedge source(fd) \in \gamma \wedge buf \in \mathbb{Z} \wedge sz \in \mathbb{Z} \\
\wedge \exists_{sf \in \theta}(sf.base \leq buf \wedge buf \leq sf.top \wedge buf + sz > sf.canary \wedge \\
\mu[sf.canary] \in SYMBOLICS \wedge is\_affine(\mu[buf, sf.canary], \delta) \wedge \\
max\_var\_index(\mu[buf, sf.canary - 1]) = n) \\
\hline
< \Sigma, \mu, \Delta, \Pi, \delta >, < \gamma, v, \sigma >, \theta \rightsquigarrow < \Sigma, \mu, \Delta, \Pi, \delta|_{\{0,...,n\}} >, < \gamma, v, \sigma >, \theta
\end{gathered}
\tag{1}
$$

Symbolic output buffer manipulation is applied when the output buffer can be influenced by some external input. Formula (2) demonstrates the operation semantics of it at the entry of write. If $buf$ or $sz$ is detected to being a symbolic value that can be influenced by external input, symbolic output buffer manipulation tries to bind these parameters to the memory area on some activation record on which the saved canary has not been overflowed yet through constraint solving. If the forged constraint $buf + sz > canary' \wedge buf \leq canary'$ is proved satisfiable, by updating the original path constraint $\Pi$, a new execution state that can lead to canary leakage is automatically generated.

$$
\begin{gathered}
\Sigma[\Delta["pc"]] = write(fd, buf, sz) \wedge source(fd) \in \gamma \wedge buf \in SYMBOLICS \\
\vee sz \in SYMBOLICS \wedge \exists_{sf \in \theta}(is\_satisfiable(buf + sz > canary' \wedge \\
buf \leq canary' \wedge canary' = sf.canary) \wedge \mu[sf.canary] \in \mathbb{Z}) \\
\hline
< \Sigma, \mu, \Delta, \Pi, \delta >, < \gamma, v, \sigma >, \theta \rightsquigarrow \\
< \Sigma, \mu, \Delta, \Pi \wedge buf + sz > canary' \wedge buf \leq canary', \delta >, < \gamma, v', \sigma >, \theta
\end{gathered}
\tag{2}
$$

Once a canary leaking state is derived, the canary leakage forger performs constraint solving on it to generate a test case that can trigger the desired canary leakage at runtime as shown in Algorithm 1, accomplishing the first step towards complete exploitability evaluation shown in Section 2.1.

---

**Algorithm 1** Canary leaking input generation

---

**Input:**
    The canary leaking state, $state :< \Sigma, \mu, \Delta, \Pi, \delta >$.
**Output:**
    The content of canary leaking input.
 1: $solver = SMTSolver()$
 2: $content = solver.solve(state.\Pi)$
 3: $out\_content = newbytes[state.\delta.count()]$
 4: **for** $i = 1$ to $state.\delta.count()$ **do**
 5:    $out\_content[i] = content[i]$
 6: **end for**
 7: return $out\_content$

---

## 5. Canary Leaking Analysis Time Exploit Generation

The canary leaking analysis time exploit generation engine consists of three components, namely canary check patcher, canary-leakage-guided fuzzing and exploitation technique scheduler. The canary check patcher first generates a patched program that is friendly to vulnerability discovery. Then, based on this patched program, the canary-leakage-guided fuzzing technique is applied generating incremental POC input that can not only leak the desired canary, but also manifest some new vulnerability afterwards. The exploitation technique scheduler then constructs an analysis time exploit based on this incremental POC input, providing a data template to complete exploitability analysis on runtime environment.

### 5.1. Canary Check Patcher

The sanity check imposed by StackGuard would generally block control-flow hijacking-related vulnerabilities including stack overflow, format string vulnerability, etc., from being discovered by current test case generation engines. The canary check patcher aims to provide a patched program that functions the same as the original binary program, except that the sanity checks would all be stripped.

The canary check patcher is implemented as a function-level analysis pass. It firstly recognizes the canary storage on the stack through the unique canary obtainment disassembly pattern shown in Figure 1. It then performs an intra-procedural analysis calculating the reaching definitions at each program point. When the reaching definition analysis is done, the canary check patcher then recognizes conditional jump instructions whose branching condition is dependent upon the stored canary. It also identifies the call instruction whose target is __stack_chk_fail_local. It then patches these instructions with 0x90, which is the NOP instruction under x86 platforms. Figure 3 demonstrates the patched version of function func in the example program shown in Figure 1a. As the sanity check no longer exists, a single stack smashing input can easily lead to the execution reaching some exploitable state. Upon the exploitable state, known exploitation techniques would be enumerated to complete the analysis time exploit generation process.

```
.text:08049225                    endbr32
.text:08049229                    lea      ecx, [esp+argc]
.text:0804922D                    and      esp, 0FFFFFFF0h
.text:08049230                    push     dword ptr [ecx-4]
.text:08049233                    push     ebp
.text:08049234                    mov      ebp, esp
.text:08049236                    push     ebx
.text:08049237                    push     ecx
.text:08049238                    sub      esp, 70h
.text:0804923B                    call     __x86_get_pc_thunk_bx
.text:08049240                    add      ebx, 2DC0h
.text:08049246                    mov      eax, large gs:14h
.text:0804924C                    mov      [ebp-0Ch], eax
.text:0804924F                    xor      eax, eax
.text:08049251                    sub      esp, 4
.text:08049254                    push     100h
.text:08049259                    lea      eax, [ebp-6Ch]
.text:0804925C                    push     eax
.text:0804925D                    push     0
.text:0804925F                    call     sub_80490A0
.text:08049264                    add      esp, 10h
.text:08049267                    sub      esp, 8
.text:0804926A                    lea      eax, [ebp-6Ch]
.text:0804926D                    push     eax
.text:0804926E                    lea      eax, (aS - 804C000h)[ebx]
.text:08049274                    push     eax
.text:08049275                    call     sub_80490B0
.text:0804927A                    add      esp, 10h
.text:0804927D                    nop
.text:0804927E                    mov      eax, [ebp-0Ch]
.text:08049281                    xor      eax, large gs:14h
.text:08049288                    nop
.text:08049289                    nop
.text:0804928A                    nop
.text:0804928B                    nop
.text:0804928C                    nop
.text:0804928D                    nop
.text:0804928E                    nop
.text:0804928F                    lea      esp, [ebp-8]
.text:08049292                    pop      ecx
.text:08049293                    pop      ebx
.text:08049294                    pop      ebp
.text:08049295                    lea      esp, [ecx-4]
.text:08049298                    retn
```

**Figure 3.** Patched function snippet. The red frame demonstrates the patched part.

### 5.2. Canary Leakage Guided Fuzzing

Canary-leakage-guided fuzzing attempts to construct some incremental POCs that can not only leak the desired canary, but also trigger some new vulnerabilities afterwards. Figure 4 demonstrates the overview of this component. Similar to coverage-guided fuzzing [22,34,35], our canary-leakage-guided fuzzing maintains a global seed queue. It keeps fetching some seed from the queue and then performing mutation on the seed, with some test cases generated. The input sources of the global seed queue include the canary leaking inputs generated in Section 4 and the test cases generated during each fuzzing iteration. Only if some seed from these sources is evaluated as promising can it be put into the global seed queue.

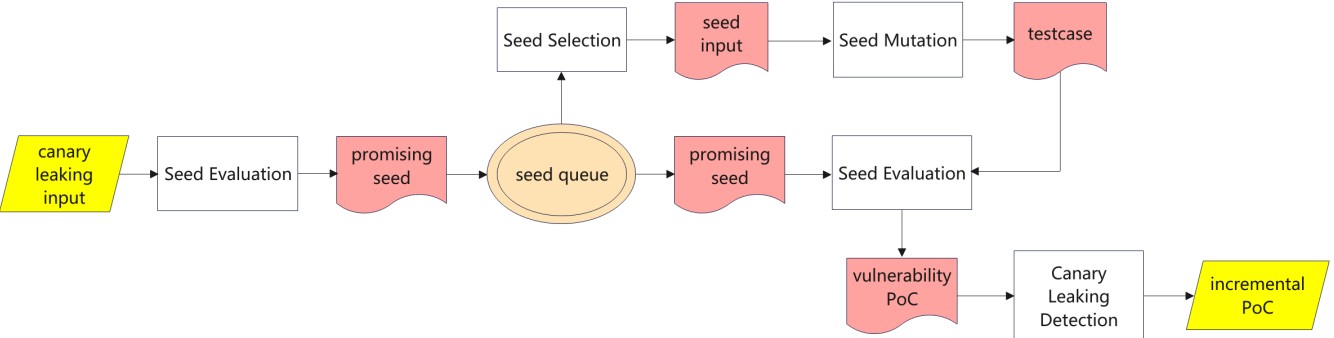

**Figure 4.** Canary-leakage-sensitive fuzzing.

Traditional coverage-guided fuzzing techniques evaluate a seed as promising only if its execution trace could hit some new control flow edge. As for canary-leakage-guided

fuzzing, we build a canary detection scheme scanning for canary value in the output streams during the dynamic execution monitor process, then integrate this scheme into the seed evaluation step, with the notion of promising seed defined as follows. If some test case can not only hit some new edge, but also dynamically leak the desired canary, this test case would then be pushed to the global seed queue. It would later be deemed as a promising seed and participate in the next test case generation iteration. In this way, we make assurance that seed inputs participating in the mutation process can effectively leak the desired canary, promoting the possibility of generating test cases implying canary leakage. We also enforce an extra execution monitor on the vulnerability-triggering test cases generated during fuzzing. Only if the vulnerability triggering test cases can exhibit canary leakage during execution can they be judged as incremental POCs. Therefore, the state space exploration is always driven towards the direction that can leak the desired canary and trigger some vulnerability afterwards, generating incremental POCs in a persistently evolutionary way.

*5.3. Exploitation Technique Scheduler*

The exploitation technique scheduler cares about five types of exploitable states, namely onStackEipOverwrite, onFuncPtrOverwrite, onFuncContentOverwrite, onStackEBPHijack and onFmtstrVul. onStackEipOverwrite denotes an execution state where the saved EIP on the stack can be controlled. onFuncPtrOverwrite denotes an execution state where some function pointer can be controlled. onFuncContentOverwrite denotes an execution state where the content some function pointer points to can be controlled. onStackEBPHijack denotes an execution state where the EBP register can be controlled. onFmtstrVul denotes an execution state where the format string parameter at some format string function call site can be controlled.

Table 2 demonstrates the relevant context that would be maintained on these states. When some exploitable state is discovered, the corresponding context information would be recorded. The exploitation technique scheduler then schedules the related techniques, generating analysis time exploit input that can exercise all the desired exploitation steps over the target vulnerability in the analysis time environment.

**Table 2.** Bound between exploitable state and exploitation techniques.

| Exploitable State | Context | Exploit Techniques |
|---|---|---|
| onStackEipOverwrite | the overwritten return pointer on stack, ESP, and sequence of symbolic bytes from ESP | direct ROP pivot to ROP on stack pivot to fake stack |
| onFuncPtrOverwrite | the content of the overwritten function pointer | point to shellcode pivot to fake stack |
| onFuncContentOverwrite | the function pointer pointing to the overwritten content and the size of the overwritten content | direct shellcode |
| onStackEBPHijack | the overwritten EBP on stack | fake EBP |
| onFmtstrVul | the format string, ESP at collage | format string control flow hijack |

## 6. Dynamic Canary-Relocation-Based Exploitability Evaluation

When an analysis time exploit input is generated, a dynamic canary-relocation-based exploitability evaluation process is adopted by CanaryExp to make the analysis time exploit fit on the targeted runtime environment. As can be seen in Figure 2, the dynamic canary-relocation-based exploitability evaluation process consists of three components, namely IO sequence analysis, canary metadata collection and dynamic canary-relocation-based exploitability session construction. This section discusses the internal working details.

### 6.1. IO Sequence Analysis

Provided with the vulnerable program $P$ and some analysis time exploit input $s0$ generated in the previous section, the IO sequence defines the order of the input and output events along the execution trace of $P$ on $s0$. As for the example program shown in Figure 1a, supposing we have some analysis time exploit $Exp_{analysis}$ containing two parts, separately one with 0x60 'A's and another with 0x60 'B's suffixed by three parts, separately a 4 byte long canary placeholder, 0xc 'B's and 0x08569181, which is the virtual address of function backdoor shown in Figure 1a, the IO sequence $ioSeq$ for $Exp_{analysis}$ is shown below:

$ioSeq$ = { $IOEntry_1$ = {type = READ, len = 0x60},
$\quad\quad\quad\quad$ $IOEntry_2$ = {type = WRITE, len = 0x6c},
$\quad\quad\quad\quad$ $IOEntry_3$ = {type = READ, len = 0x74},
$\quad\quad\quad\quad$ $IOEntry_4$ = {type = WRITE, len = 0x74}
}

Resorting to the function level instrumentation utilities shown in Table 1, we hook all the input and output relevant API functions, making a just-in-time record for each input/output event. When the execution terminates, the complete IO sequence is derived.

### 6.2. Canary Leakage Meta Data Collection

The canary metadata collection component aims to record information on the desired canary that is necessary to extract the canary from the output stream and locate the canary-related field in the input stream. More specifically, three types of information would be gathered: separate canary leaking offset, canary leaking format and canary relocation offsets.

The canary leaking offset and canary leaking format provide the information necessary to extract canary content from a given output stream. These two types of information are collected during the output analysis of the execution trace. As to canary relocation offsets, we perform concolic execution along the execution trace of the analysis time exploit on the target program, recording the stack locations holding the runtime canary. When the execution finishes, the memory content held on the bookmarked locations is checked. If some of the content remains symbolic, the specific input offsets would be extracted from the corresponding symbolic value, providing support for canary runtime relocation that would be discussed in discussed in Section 6.3.

### 6.3. Canary-Relocation-Based Exploitation Session Construction

With the IO sequence and canary leakage metadata in mind, the canary-relocation-based exploitation session construction component makes just-in-time transformation on the analysis time exploit during the dynamic exploitation process, evaluating the exploitability of the target vulnerability in a completely automatic way.

Algorithm 2 presents the inner working details. Provided with the communication channel *proc* on the vulnerable program, the analysis time exploit $Exp_{analysis}$, the word size *ptrSz* of the target processor, the IO sequence *ioSeq* and the canary metadata *canaryMd*, the algorithm simulates input and output events sequentially in the order specified by *IOSeq*.

When some READ event should be simulated, the relevant part in *content* is determined and then sent to the vulnerable program via *proc.send* (line 6–9). When some WRITE event should be simulated, we fetch the output data of the vulnerable program via *proc.receive*, and then check if the desired canary exists in the output stream (lines 11–12). If it is confirmed existing, the actually leaked canary is calculated at line 13. This leaked canary is then utilized to relocate the input segment at the offset specified by *canaryMd.offs* at lines 14–16. The updated exploit would then be sent to the target in the following READ session, completing the vulnerability exploitability evaluation process in a canary sanity check bypassing way.

---

**Algorithm 2** Runtime exploitation session with dynamic canary relocation

---

**Input:**

   The full duplex channel to connect to the target process, *proc*.

   The analysis time exploit input, $Exp_{analysis}$.

   The word size of the architecture of the target program, *ptrSz*. 4 for 32bit executables, 8 for 64 bit executables.

   The IO sequence, $ioSeq = \{IOEntry_1, IOEntry_2, \ldots, IOEntry_n\}$, where $IOEntry_i$ $(1 \leq i \leq n)$ is the record describing the $i$th IO action.

   The canary metadata, $canaryMd = <loff, lfmt, offs>$, where $loff$ and $lfmt$ denotes the offset and format of the leaked canary in the output stream, $offs = \{off_1, off_2, \ldots, off_n\}$ denotes the fields in the input stream that should be fixed with the leaked canary.

**Output:**

   $contents = Exp_{analysis}$
   $si = 0$
   $ri = 0$
   $canary = 0$
   **for** each $IOEntry \in ioSeq$ **do**
      **if** $IOEntry.type == READ$ **then**
         $si\_end = si + IOEntry.len$
         $proc$.send($contents[si : si\_end]$)
         $si = si\_end$
      **else if** $IOEntry.type == WRITE$ **then**
         $recvContent = proc.receive(IOEntry.len)$
         **if** $ri <= canaryMd.loff$ $AND$ $canaryMd.loff <= ri + IOEntry.len$ **then**
            $canary = extractCanary(recvContent, canaryMd.loff - ri,$
                           $canaryMd.lfmt)$
            **for** each $off$ in $canaryMd.offs$ **do**
               $contents[off, off + ptrSz] = canary$
            **end for**
         **end if**
         $ri = ri + IOEntry.len$
      **end if**
   **end for**

---

## 7. Evaluation

We implemented a prototype of CanaryExp based on the binary analysis platform angr [28], well-known fuzzer AFL++ [29], whole-system symbolic executor S2E [30], and the exploit development library pwntools [36].

The canary leakage POC generation is implemented on S2E with 2342 lines of C++ code. As for canary leaking analysis time exploit generation, there are 154 lines of Python code on the angr side implementing the canary check patcher, 4814 lines of C code on AFL++ implementing canary-leakage-guided fuzzing, 3445 lines of C++ code implementing the exploitation technique scheduler on S2E. For dynamic canary-relocation-based exploitability evaluation, the IO sequence analysis and canary metadata collection are implemented on S2E, consisting of 1546 lines of C++ code. The exploitability session construction process is implemented on pwntools, consisting of 564 lines of Python code.

In this section, we present the evaluation results of this system. These experiments are all carried out in a 64-bit Ubuntu 22.04.3 LTS system on a workstation with 128 G RAM and Intel(R) Core(TM) i9-12950HX CPU @ 2.5 GHz. These experiments are conducted to answer the following questions:

- RQ1: Can CanaryExp generate canary leaking input and incremental POC input samples more efficiently than current test case generation techniques?

- RQ2: Can CanaryExp generate an exploitation session that can automatically evaluate the exploitability of the vulnerable programs with the StackGuard protection mechanism effectively bypassed?

*7.1. Benchmarks*

The benchmarks we constructed to evaluate our system are shown in Table 3. These benchmarks are composed of six test programs, eight challenges from CTF events and four real-world applications with exposed vulnerabilities. These benchmarks are all user-mode applications running on the Linux platform in the ELF format. These programs are collected based on the following criteria.

- Vulnerabilities commonly existing in binary programs should be covered by the benchmarks. As shown in Table 3, vulnerabilities including stack overflow, integer overflow, format string, use after free, and heap overflow are all covered in this collection. These are all common vulnerabilities exhibited in binary programs.
- The benchmark programs should be protected by StackGuard, and forging a canary leakage event is a must in exploitability evaluation. As shown in Table 3, the 'Leak Type' column lists the method that should be taken to forge the desired canary leakage event. Note that 'ARR' and 'SBM' separately stand for activation record restoration and symbolic output buffer manipulation discussed in Section 4.2.
- The benchmarks should also cover as much as possible the exploitation mitigation techniques. As we can see, in the 'protection' column in Table 3, except for StackGuard, mainstream protection mechanisms including NX and ASLR are all covered in this collection. It is worth noting that for benchmark programs that run under an ASLR environment, we complete the exploitability evaluation process by merging CanaryExp with our previous work [37].

**Table 3.** Evaluation benchmark.

| Dataset | Program | Vulnerability Type | Protections | Leak Type |
|---|---|---|---|---|
| test programs | stack1 | stack overflow | Canary, NX, ASLR | ARR |
| | intovf | stack overflow, integer overflow | Canary, NX | ARR |
| | fmtstr1 | format string | Canary | SBM |
| | fmtstr2 | format string | Canary | SBM |
| | heap-vul1 | use after free | Canary | SBM |
| | heap-vul2 | heap overflow | Canary, NX | ARR |
| bjdctf 2020 | babyrop2 | stack overflow, format string | Canary, NX | SBM |
| moectf 2020 | baby_canary | stack overflow, format string | Canary, NX, ASLR | SBM |
| BugkuCTF | canary | stack overflow | Canary, NX | ARR |
| insomnihack CTF 2016 | microwave | stack overflow, format string | Canary, NX, ASLR | SBM |
| 2018 anheng cup CTF | babypie | stack overflow | Canary, NX, ASLR | ARR |
| CSAW Quals CTF 2017 | scv | stack overflow | Canary, NX | SBM |
| xctf | Mary_Morton | stack overflow, format string | Canary, NX | SBM |
| | babystack | stack over flow | Canary, NX | ARR |
| real applications | proftpd 1.3.1 (CVE-2006-6563) | stack overflow | Canary, ASLR | ARR |
| | iwconfig v26 (BID-8901) | stack overflow | Canary, NX | ARR |
| | dnsmasq 2.77 (CVE-2017-14493) | stack overflow | Canary, NX, ASLR | ARR |
| | rsync 2.5.7 (CVE-2004-2093) | stack overflow | Canary, NX | ARR |

*7.2. Effectiveness in Canary Leaking Input and Incremental POC Input Generation*

To answer RQ1, we compare CanaryExp against well-known engines including AFL, AFL++, S2E and QSYM. To guarantee computation fairness, We allocate the same quota on resources for computation, which is composed of two logical processors, 64 GB physical memory and a 2 h time quota. Note that we set the logical processor count to 2 because CanaryExp takes exactly two logical processors during calculation–one for canary leaking input generation and the other canary leakage persistent POC input generation. We list the specific configurations for each engine in Table 4.

**Table 4.** Engine configuration.

| Engine | Configuration Setup |
| --- | --- |
| CanaryExp | 1 instance of AFL++ slave mode, 1 instance of S2E |
| AFL | 1 instance for AFL master mode, 1 instance for AFL slave mode |
| AFL++ | 1 instance for AFL++ master mode, 1 instance for AFL++ slave mode |
| S2E | 2 instances of S2E, working in parallel |
| QSYM | 1 instance of a PIN-based concolic executor, 1 instance for AFL slave mode |

We conduct two experiments as follows. To compare CanaryExp with other test case generation engines on canary leaking input generation, we launch them on the same initial input corpus containing POC samples that can lead the execution trigger some vulnerability, with an introspection process launched at the same time checking whether the generated test cases can leak the desired canary, with the time cost recorded at the same time. To compare CanaryExp with other test case generation engines on incremental POC input generation, we launch them on the same initial input corpus containing canary leakage samples, and provide the sanity check stripped programs instead of the original StackGuard-protected versions. We also calculate the time cost in generation through dynamic monitoring of the engines' output at the same time.

Table 5 shows the results. We can see that CanaryExp is able to generate canary leaking input at low time cost while other engines generally fail in the given time quota. The root cause of this, we believe, lies in the uniquely proposed canary leakage forger CanaryExp. Through introspecting upon the output stream, this component dynamically transforms some execution state into a canary leaking state when identifying the output context satisfying the necessary transformation condition, therefore establishing the advantage for CanaryExp in this facet over other known test case generation engines.

We can also conclude that CanaryExp is able to generate incremental POC samples for the benchmark programs more effectively than other engines. We believe this is due to CanaryExp's unique canary sensitivity. As for coverage-guided fuzzing engines like AFL and AFL++, they generally distinguish test cases by their branch coverage feedbacks. Under this paradigm, a specific branch coverage map generally groups an equivalent set of test cases, with the first test case exhibiting this branch coverage selected as the representative; other test cases subsequently generated following the same branch coverage would simply discarded in future test case generation processes. However, this is not the case for canary leaking input generation. As we can see from Figure 1, the path taken by the canary leakage sample can be easily reached through simple fuzzing. Therefore, though carrying evident canary leakage feature, the canary leaking input would simply not be deemed as interesting and discarded immediately as some cheap representative already exists.

For S2E and QSym, the symbolic executor part would also ignore the canary leaking context information during path space exploration. Even though the input corpus provided to S2E and QSym already implies some canary leakage event, they only enforce blind path exploration obtaining poor efficiency in incremental POC generation. CanaryExp,

however, because of its unique sensitivity on canary leakage events, can definitely forge the necessary canary leakage event in canary leaking input generation and identify the interesting and critical step carried out by the canary leakage sample in incremental POC generation, therefore obtaining better results in this criterion.

**Table 5.** Time cost of canary leaking input generation and incremental POC generation.

| Program | Canary Leaking Input Generation Time (s) | | | | | Incremental POC Generation Time (s) | | | | |
|---|---|---|---|---|---|---|---|---|---|---|
| | CanaryExp | AFL | AFL++ | S2E | QSYM | CanaryExp | AFL | AFL++ | S2E | QSYM |
| stack1 | 141 | - | - | - | - | 414 | 906 | 1034 | 2135 | 923 |
| intovf | 158 | - | - | - | - | 453 | 843 | 936 | 1785 | 793 |
| fmtstr1 | 276 | - | - | - | - | 436 | 903 | 857 | 1890 | 917 |
| fmtstr2 | 303 | - | - | - | - | 321 | 878 | 745 | 1905 | 671 |
| heap-vul1 | 341 | - | - | - | - | 509 | 862 | 913 | 2413 | 874 |
| heap-vul2 | 323 | - | - | - | - | 537 | 1047 | 1132 | 2321 | 908 |
| babyrop2 | 415 | - | - | - | - | 424 | 764 | 613 | 1621 | 598 |
| baby_canary | 421 | - | - | - | - | 383 | 601 | 596 | 1422 | 523 |
| canary | 517 | - | - | - | - | 395 | 503 | 497 | 874 | 488 |
| microwave | 523 | - | - | - | - | 374 | 465 | 544 | 916 | 421 |
| babypie | 325 | - | - | - | - | 344 | 541 | 566 | 1212 | 497 |
| scv | 443 | - | - | - | - | 284 | 412 | 445 | 842 | 372 |
| Mary_Morton | 513 | - | - | - | - | 245 | 378 | 332 | 674 | 405 |
| babystack | 414 | - | - | - | - | 304 | 417 | 456 | 786 | 433 |
| proftpd 1.3.1 | 516 | - | - | - | - | 483 | 561 | 577 | 1231 | 584 |
| iwconfig v26 | 464 | - | - | - | - | 421 | 514 | 498 | 1314 | 501 |
| dnsmasq 2.77 | 515 | - | - | - | - | 547 | 786 | 775 | 1874 | 694 |
| rsync 2.5.7 | 432 | - | - | - | - | 442 | 678 | 661 | 1143 | 578 |

### 7.3. Effectiveness in Exploit Generation

To answer RQ2, we compare CanaryExp with REX [3], a well-known AEG system developed by the Shellphish team. We provide the same initial seed corpus that contains input samples that can leak the canary at runtime to the two engines, test if they can generate analysis time exploit and complete the exploitation on the vulnerable programs. Note that we use the canary check stripped version of the benchmark programs as a test suite in experiments on analysis time exploit generation and the original version of the benchmark programs as a test suite in runtime exploitation session.

Table 6 shows the results. We can see that although REX can generate analysis time exploit successfully, as CanaryExp proved with the patched programs. As shown in the column 'Exploit Generation' depicted in Table 6, it generally fails in exploitability evaluation on the original versions of the benchmarks, while CanaryExp, on the other hand, can finish the tasks, as depicted in column 'PWN ability' in Table 6. We believe the novel runtime canary relocation technique we bring out in this paper should account for this difference. As the runtime canary relocation technique can effectively bridge the difference between the analysis time canary and runtime canary, CanaryExp can bypass the sanity check imposed by StackGuard in a completely automatic way, gaining superiority on this issue over REX.

**Table 6.** Experimental results of exploit generation.

| Program | Engine | Exploit Generation | PWN Ability |
|---|---|---|---|
| stack1 | CanaryExp | ok | success |
| | REX | ok | fail |
| intovf | CanaryExp | ok | success |
| | REX | ok | fail |
| fmt1 | CanaryExp | ok | success |
| | REX | ok | fail |
| fmt2 | CanaryExp | ok | success |
| | REX | ok | fail |
| heap1 | CanaryExp | ok | success |
| | REX | ok | fail |
| heap2 | CanaryExp | ok | success |
| | REX | ok | fail |
| babyrop2 | CanaryExp | ok | success |
| | REX | ok | fail |
| baby_canary | CanaryExp | ok | success |
| | REX | ok | fail |
| canary | CanaryExp | ok | success |
| | REX | ok | fail |
| microwave | CanaryExp | ok | success |
| | REX | ok | fail |
| babypie | CanaryExp | ok | success |
| | REX | ok | fail |
| scv | CanaryExp | ok | success |
| | REX | ok | fail |
| Mary_Morton | babyrop2 | ok | success |
| | REX | ok | fail |
| babystack | CanaryExp | ok | success |
| | REX | ok | fail |
| proftpd 1.3.1 | CanaryExp | ok | success |
| | REX | ok | fail |
| iwconfig v26 | CanaryExp | ok | success |
| | REX | ok | fail |
| dnsmasq 2.77 | CanaryExp | ok | success |
| | REX | ok | fail |
| rsync 2.5.7 | CanaryExp | ok | success |
| | REX | ok | fail |

## 8. Discussion

This paper presents CanaryExp, an automatic exploitability evaluation scheme for vulnerable binary programs that are protected by StackGuard. To solve the canary bypassing issue generally ignored by current AEG systems, CanaryExp employs three novel techniques: separate canary leakage POC generation, canary leakage sensitive fuzzing and dynamic canary relocation.

The canary leakage POC generation mechanism monitors the execution trace along a vulnerability triggering input, with some canary leaking state intentionally forged through

output analysis. Upon the constructed canary leaking state, some canary leaking samples are derived.

The canary leaking analysis time exploit generation takes the canary leaking samples generated above as input, progressively pushing forward the multi-step exploitation automaton by firstly generating some incremental POC that can trigger some new vulnerability after leaking the runtime canary, then constructing an analysis time exploit that can not only leak the canary but also complete the exploitation in analysis time environment.

The dynamic canary-relocation-based exploitability evaluation firstly collects metadata related to the desired canary, then with this information and the analysis time exploit in mind, constructs the exploitation session that can adapt to the runtime environment by intelligently leaking the canary and performing automatic relocation on the input exploit stream.

As shown in Section 7, CanaryExp performs better in canary leakage test case generation, proven an effective solution in the automatic exploitation of vulnerable programs protected by StackGuard. However, there is still some room for improvement for Canary-Exp, as discussed below.

- **Automatic Exploitable Heap Layout Generation.** Automatic generating exploitable heap layout is the most critical issue when evaluating the exploitability of heap-related vulnerabilities. Though some pattern-based solutions including [11,13] have been proposed working as a per-state evaluation method in the blind exploration process on the path space of the target program, it is necessary to devise some directed analysis to improve the efficiency in exploit generation for heap vulnerabilities.
- **Support on Multi-Threading Vulnerabilities.** Current AEG systems generally calculate the semantics of program instructions in sequential execution order. However, this does not fit for case when the vulnerability arises only in multi-threading environments, e.g., concurrency vulnerabilities. As CanaryExp currently also adopts the same sequential calculation model, a multi-threading-environment-oriented calculation model is desired to make it applicable in exploitability evaluation of vulnerabilities in multi-threading environment.
- **AEG on interpreters.** Current AEG systems generally depend heavily on symbolic execution to reason about the internal relationship between program variables and external input. CanaryExp also fails in this case. However, for interpreter like programs, as they often translate the input program into some abstract syntax tree structure and then generate just-in-time code on it, the symbolic variables introduced by the original input program may get lost due to this intermediate translation, directly leading current AEG methods fail in this case. Some intermediate layer should also be introduced to make AEG adaptable regarding this issue.

## 9. Related Work

### 9.1. Feedback Guided Fuzzing

Feedback-guided fuzzing is currently the most prevalent test case generation technique. By instrumenting the target programs through techniques including compilation [22,35], dynamic binary translation [29,31], dynamic binary instrumentation [38], binary rewriting [39] or collecting the execution statistics through advanced processor provided features [40], many fuzzers collect different kinds of feedback information during the execution phase of the generated test cases. If some new feedback is exhibited, the corresponding test case can then be deemed as interesting and then participate in the next test case generation iteration. In this way, the feedback-guided fuzzing process is also known as evolutionary fuzzing.

Branch coverage is the classic feedback originally defined by fuzzers including AFL [22], honggfuzz [35] and AFL++ [29]. By collecting the hit count of each branch instruction of the target program, these fuzzers keep discovering new test cases in which to can exercise some new branch instructions. LOLLY [41] defines state sequence as feedback in which a novel sequence guided directed fuzzing technique is proposed keeping state exploration

towards user-specified program statements. aflfast [42] and GREYONE [43] define taint of variables as feedback so as to perform data flow sensitive fuzzing.

### 9.2. Automatic Exploit Generation

Brumley et al. [44] proposed a patch-based solution for programs generating exploit input with the aid of vulnerability patches. Thanassis Avgerinos et al. proposed AEG [1], generating exploit inputs through the synthesis of different symbolic execution techniques. As the subsequent production over AEG, MAYHEM [4] was proposed in 2012, generating exploit inputs through interleaving symbolic execution and concolic execution. Based on S2E [30], a whole-system symbolic executor for binary programs, Huang et al. proposed CRAX [2] that generates exploits for stack overflow vulnerability through automatic derivation of exploitable state through concolic execution and utilizing constraint solving upon the crafted state. Revery [45] was proposed in 2018, generating exploits in a state-stitching way.

As for heap-related vulnerabilities, many solutions have also been proposed nowadays. RELAY [11] was proposed in 2020, capable of evaluating exploitability on metadata corruption vulnerabilities. HAEPG [13] was also proposed in 2020, simulating known exploitation techniques to facilitate automatic exploitation for heap vulnerabilities under glibc platforms. Maze [8] was proposed in 2021, which implements automated heap layout manipulation through modeling heap feng shui techniques.

However, currently, few solutions discuss automatic exploitation under StackGuard conditions. LAEG [46], to our knowledge, proposed the first solution on AEG for StackGuard-protected programs. However, this method works under a strict assumption that the initial POC input provided to the system implies some uninitialized data vulnerability that can directly leak the critical information. In other words, it focuses only on the back end of this problem, without any canary leakage primitive construction attempts, imposing strict requirements on the initial POC input. Our method, however, because of the active canary leakage forge and dynamic canary relocation mechanisms it adopts, can leak and fix the desired canary all by itself, providing automatic AEG workflow much more complete than LAEG.

### 10. Conclusions

We present CanaryExp, a novel vulnerability exploitability evaluation solution for programs that are protected by StackGuard. CanaryExp works in a multi-step exploitation paradigm. It first generates test cases that can exhibit the desired canary leakage event through a canary leakage POC generation process. Then, based on the generated address leakage input, a canary leaking analysis time exploit generation method is proposed, generating an exploit input that can fit the exploitation process in the analysis time execution environment. Based on the analysis time exploit input, a dynamic canary-relocation-based exploitability evaluation scheme is devised, providing a completely automated exploitability evaluation pipeline in a canary-sensitive way. The results of our experiments show that CanaryExp can perform better than current AEG solutions in both canary leakage sample generation and runtime canary adaption.

However, CanaryExp still has some limitations. It cannot derive an exploitable heap layout automatically from scratch. It also fails to analyze vulnerabilities existing in multi-threading environments and interpreter-like programs. These supports are planned to be added in the future to further enhance CanaryExp's scalability and capability in real-world applications.

**Author Contributions:** Conceptualization, H.H., Y.L. and K.Z.; methodology, H.H. and K.Z.; software, H.H. and K.Z.; validation, H.H.; investigation, H.H.; resources, Y.L. and J.Z.; writing—original draft preparation, H.H.; writing—review and editing, Y.L., K.Z. and J.Z.; supervision, H.H. and K.Z.; project administration, Y.L. All authors have read and agreed to the published version of the manuscript.

**Funding:** This research received no external funding.

**Institutional Review Board Statement:** Not applicable.

**Informed Consent Statement:** Not applicable.

**Data Availability Statement:** Data are available on request to the authors. The data are not publicly available due to privacy.

**Acknowledgments:** We would like to sincerely thank all the reviewers for your time and expertise on this paper. Your insightful comments help us improve this work.

**Conflicts of Interest:** The authors declare no conflict of interest.

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
