# Peer review of "CanaryExp: A Canary-Sensitive Automatic Exploitability Evaluation Solution for Vulnerabilities in Binary Programs"

_applsci, doi:10.3390/app132312556_

Round 1

Reviewer 1 Report

Comments and Suggestions for Authors

1. The critical literature review was well prepared by the authors of the article.

2. The bibilographic items included in the literature list were well prepared, but items from one country predominate.

3. In the list of references, please explain or add the use of item no. 21, add the date of access to item no. 4, and in item 35 the number of pages or pp. 29, or (1-27).

4. The conclusions of the entire article are very poorly developed - only three sentences (the second sentence is too long). In your conclusions, please describe the research, results and the purpose of further research.

5. Description of drawing no. 3, the beginning of the sentence should be with a capital letter.

6. Fig. 1 and 2 are difficult to read, especially subsection no. c in fig. 1 (larger font), the same for section no. d (description in red font) and the table. In Fig. 2, increase the font size or use Bold.

7. To better understand the formula (expression), e.g. no. 1, it is worth adding a list of the symbols used - e.g. pi. sum, etc.

8. Good article, but requires some minor corrections.

Author Response

Dear reviewer,

I have read your review report and finished the revisions according to your suggestions.

As for point 2, the bibliographic items are adjusted.

As for point 3, the use of item no.21 is explained. The date of access to item no.4 is added. Item 35 is also fixed.

As for point 4, the conclusions are rewritten according to your suggestion.

As for point 5, drawing no.3 is adjusted.

As for point 6, figure 1 and figure 2 are redrawn.

Sincerely thank you for your suggestion and best wishes !

Yours

Reviewer 2 Report

Comments and Suggestions for Authors

The context of the research is clearly described and is appropriate to the topic of the research. Basic international bibliographical references are provided in the area of system security and vulnerability.

The general research objective is not clearly defined, it would be advisable to include it both in the body of the document and in the summary. It is also advisable to include the research questions raised based on the proposed model.

The methodological and procedural development of the CanaryExp is perfectly founded, explained and justified. Practical examples of its standard are provided that justify its use within the context of application. The evaluation of the program appears adequate and guarantees its application in a justified manner.

The conclusions point out the most notable findings regarding the use of the CanaryExp. However, it is highly recommended to include a space for the limitations of the research, since their mention is very rare and it is advisable not only to provide the benefits and advantages of CanaryExp, but also to include its problems or negative aspects, along with future lines. of research that derive from this work, since they are not referenced in the text.

Author Response

Dear reviewer,

I have read your review report and finished the revisions according to your suggestions.

As you stated that the general research objective is not clearly defined, I want to humbly  point that the research objective has been stated in the introduction section. The conclusion is rewritten according to your suggestion.

As for the negative aspects, the limitation of this work is discussed mainly in section 8. The defects and future lines are also manifested in the conclusion part.

Sincerely thank you for your suggestion and best wishes !

Yours

Reviewer 3 Report

Comments and Suggestions for Authors

The submission presents a utility called CanaryExp that automates the identification of exploitable vulnerabilities in binary programs in ELF format. The submission illustrates the design of CanaryExp and discusses the workings of each of the macro-components of the internal organization of that utility. The submission also exhibits experimental results that provide evidence of their tool doing better than other tools in the state of the art.

If we were to stop at this level, the submission would have all of the required qualities to pass the bar of acceptance.

Unfortunately, that is not the case, as the submission has very poor quality in the explanations that it provides, which often cause more obfuscation than clarity to the reader. The attached file highlights cases of obfuscation, where the text is almost completely incomprehensible or omissive. In fact, this defect has two parts: (1) language quality that is poorly attended, where horrendously long sentences are provided, which cannot be sensibly parsed by the reader (and presumably by the writers too); (2) omission of detail at key points, where the authors do not explain what they mean, do not back up their claims, do not provide sufficient details. In multiple instances, it is evident that the authors do not care for the reader, and expect too much to be "guessed" or read in-between-lines (as in jargon, implicit domain habits, and other such occurrences).

The submission also has another important defect in structure: (a) they go deep into presenting their solution, without setting sufficient ground to demonstrate that the problem they want to solve is indeed open in the state of the art; (b) they pose research questions that are very vertical and make only sense in relation to their own proposed utility, which however cannot be considered a scientific contribution in the face of defect (a); (c) they run comparative experiments with tools chosen by the authors themselves without demonstrating that they are "the state of the art"; (d) they discuss related works at the very end of the paper, when that discussion has little practical value.

On the whole, the work presented by the authors has potential, but the submission as it stands should be rejected because it fails in a large number of required quality traits. Fixing the reported defects would require rewriting the submission profoundly, going into much more explanatory depth than it currently does, and reorganizing its flow of argument.

Comments on the Quality of English Language

The attached document highlights numerous occurrences of grammatical errors in the manuscript, with recommendations or suggestions for fixing.

Author Response

Dear reviewer,

I have read your review report and made a complete revision on my script. The revisions are made according to every suggestion you provide in your comments. Sincerely thank you for your suggestions. Your suggestions are really beneficial for me. 

Best wishes.

Yours